# Feasibility of a wearable inertial sensor to assess motor complications and treatment in Parkinson's disease

Nuria Caballol[1,2]*, Àngels Bayés[1], Anna Prats[1,3], Montserrat Martín-Baranera[4,5‡], Paola Quispe[1‡]

1 Movement disorders Unit, "UParkinson", Centro Médico Teknon, Grupo Hospitalario Quirón, Barcelona, Spain, 2 Neurology Department, Movement Disorders Unit, Complex Hospitalari Moisès Broggi, Consorci Sanitari Integral, Sant Joan Despí, Barcelona, Spain, 3 Hospital Universitari Germans Trias i Pujol, Barcelona, Spain, 4 Clinical Epidemiology and Research Department, Complex Hospitalari Moisès Broggi, Consorci Sanitari Integral, Sant Joan Despí, Barcelona, Spain, 5 Universitat Autònoma de Barcelona, Barcelona, Spain

☯ These authors contributed equally to this work.
‡ These authors also contributed equally to this work.
* nuriacaballol@hotmail.com

**Data Availability Statement:** All relevant data are available upon request, due to ethical restrictions imposed by Spanish Agency for Drugs and Medical Devices and the Local Clinical Research Ethics

## Abstract

### Background

Wearable sensors-based systems have emerged as a potential tool to continuously monitor Parkinson's Disease (PD) motor features in free-living environments.

### Objectives

To analyse the responsivity of wearable inertial sensor (WIS) measures (On/Off-Time, dyskinesia, freezing of gait (FoG) and gait parameters) after treatment adjustments. We also aim to study the ability of the sensor in the detection of MF, dyskinesia, FoG and the percentage of Off-Time, under ambulatory conditions of use.

### Methods

We conducted an observational, open-label study. PD patients wore a validated WIS (STAT-ON™) for one week (before treatment), and one week, three months after therapeutic changes. The patients were analyzed into two groups according to whether treatment changes had been indicated or not.

### Results

Thirty-nine PD patients were included in the study (PD duration 8 ± 3.5 years). Treatment changes were made in 29 patients (85%). When comparing the two groups (treatment intervention vs no intervention), the WIS detected significant changes in the mean percentage of Off-Time (p = 0.007), the mean percentage of On-Time (p = 0.002), the number of steps (p = 0.008) and the gait fluidity (p = 0.004). The mean percentage of Off-Time among the patients who decreased their Off-Time (79% of patients) was -7.54 ± 5.26. The mean

Committee, related to approved consent procedure and protecting privacy. Data can be available upon request to qs.investigacion@quironsalud.es.

**Funding:** The author(s) received no specific funding for this work.

**Competing interests:** I have read the journal's policy and the authors of this manuscript have the following competing interests: AB is shareholder of Sense4Care. This does not alter our adherence to PLOS ONE policies on sharing data and materials. AB reports receiving honoraria from Bial and Zambon. NC reports receiving consultancy fees as member of the advisory board from Italfármaco, Abbvie and Zambon, speaking fees from Lundbeck and Zambon, and honoraria from Bial, Zambon and UCB. AP declares no conflict of interest. PQ declares no conflict of interest. This does not alter our adherence to PLOS ONE policies on sharing data and materials.

percentage of On-Time among the patients that increased their On-Time (59% of patients) was 8.9 ± 6.46. The Spearman correlation between the mean fluidity of the stride and the UPDRS-III- Factor I was 0.6 (p = <0.001). The system detected motor fluctuations (MF) in thirty-seven patients (95%), whilst dyskinesia and FoG were detected in fifteen (41%), and nine PD patients (23%), respectively. However, the kappa agreement analysis between the UPDRS-IV/clinical interview and the sensor was 0.089 for MF, 0.318 for dyskinesia and 0.481 for FoG.

## Conclusions

It's feasible to use this sensor for monitoring PD treatment under ambulatory conditions. This system could serve as a complementary tool to assess PD motor complications and treatment adjustments, although more studies are required.

## Introduction

Parkinson's disease (PD) is a complex neurodegenerative disorder characterized by a wide range of motor and non-motor symptoms. Dopaminergic treatments can improve symptoms and quality of life. However, motor fluctuations (MF), non-motor fluctuations (NMF), dyskinesia and freezing of gait (FoG) complicate the management of PD [1–3]. Accurate identification of motor symptoms, motor fluctuations and Off-Time is crucial to making a precise treatment adjustment. The clinical interview, a set of recommended clinical scales and patient's diaries are widely used for this purpose [4–7]. Nonetheless, the real time for each patient in the setting of the clinical practice is sometimes limited and not all the symptoms can be captured properly. Besides, intra and interrater correlations of UPDRS are low and give only a "quick snap picture" of the motor state of the patient [8]. Completing patient's diaries require some effort, but they are completed without difficulty in most instances [9]. However, some patients may struggle to register motor complications (MC) accurately. Recall bias, reduced diaries compliance and diary fatigue can also be common [10,11].

New technologies and wearable sensors-based systems have emerged in the last decade to objectively assess motor PD symptoms for long periods in free-living environments [12–19]. Novel devices based either on machine-learning approaches or statistical-based methodology, can be used to capture and monitor motor PD symptoms and MC, or to objectively assess response to dopaminergic therapy [20–33]. The Kinesia 360^TM is composed of two sensors, wrist-worn and ankle-worn [32]. It uses a gyroscope, and the algorithms give information regarding tremor, dyskinesia, slowness, mobility, posture and steps. The PKG^TM has been extensively used. It is a waist-worn sensor that can detects ON/OFF states, bradykinesia, dyskinesia, tremor, and inactivity state but not FoG, gait or falls [26–30]. The PDMonitor^TM uses a five-device system and can detect ON/OFF MF, bradykinesia, dyskinesia, tremor, FoG, gait measures and rest state but not falls. Although these novel tools are promising, some challenges such as the lack of external validation could limit their implementation [8]. The main challenge relies on comparing the clinical practice with devices against the clinical practice without devices and seeing if the wearable sensor-based systems can improve PD motor symptoms. So far, only comparisons between sensors and questionnaires have been done, but no clear evidence has been shown yet.

The STAT-ON[TM] is a wearable inertial sensor (WIS) that was certified as a medical device in June 2019. The system provides numerical and graphical information of the motor symptoms associated with PD. The STAT-ON reports include the percentages of Off-Time, On-Time, dyskinesia, and FoG (Fig 1A and 1C). The WIS can capture On, Off or intermediate states (Fig 1B). When no movement is detected, the report shows this period in grey colour (Fig 1B). The sensor also captures the gait fluidity (GF) (Fig 1D). The minutes walked, the number of steps per day, the cadence and the number of falls can also be detected.

A recent study that analysed the opinion of different PD experts in Spain regarding WIS clinical utility showed that the system was considered "quite" to "very useful" by 74% of the neurologists [34]. Moreover, a single-blind, multicentre, randomised controlled clinical trial comparing the effectiveness of the system with other clinical monitoring methods is being conducted [35]. Satisfaction and usability of the system among PD patients have been also analysed using the Quebec User Evaluation of Satisfaction with assistive Technology (QUEST) questionnaire and the System Usability Scale (SUS) questionnaire respectively [36–38]. The level of satisfaction of the device was high according to QUEST scale (all items scored between 4 "quite satisfied" and 5 "very satisfied"), except for the item "easy in adjusting" (47%), which had a lower score [38].

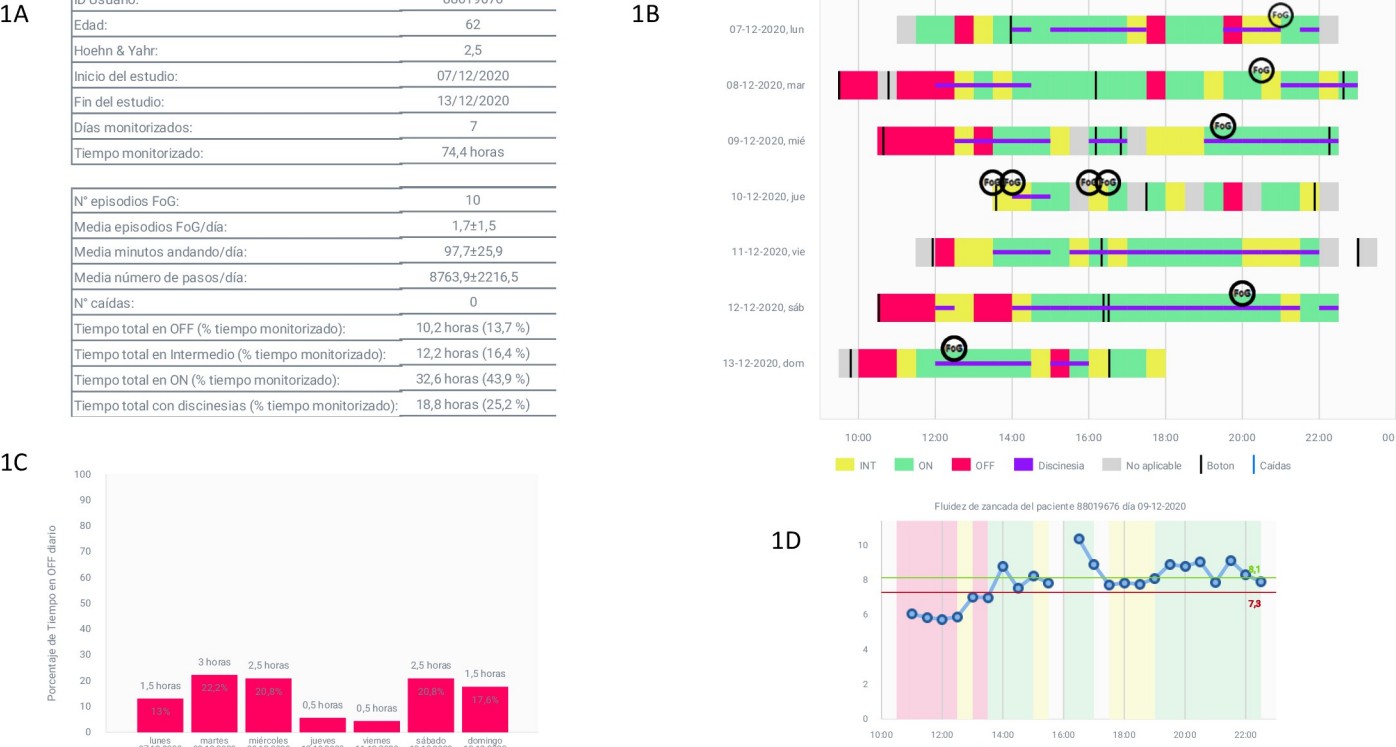

**Fig 1. Examples of sensor patients' results. A**. Example of the first page of the STAT-ON[TM] report. It contains information regarding the total time of monitorization along the 7-day-period. **B.** Example of the report that is shared with the patient. Color legend. Green: On state; Red: Off state; Yellow: Intermediate state; Grey: Inactivity or rest state. No walking detection; Freezing of Gait (FoG) is represented by a black round; Horizontal purple line: Dyskinesia; Vertical black line: Button event (in the present study, corresponds with the levodopa dose); Vertical blue line: Fall. **C**. The figure shows the total time and the percentage of Off state of every day. The height of each bar depends on the percentage of Off-Time per day. **D**. This figure shows the GF along the day of the third day of monitoring of the same patient of the STAT-ON[TM] report above. The GF is worse in the morning.

Regarding the sensitivity and specificity of the algorithms embedded into the sensor for detecting On-Off fluctuations, dyskinesia or FoG, a set of studies have demonstrated that both are around 0.9, with few variations depending on the algorithm [39–47]. However, there has been no published data regarding the responsivity of the sensor's measures after treatment adjustments. Besides, and to the best of our knowledge there are no studies analysing the agreement of the detection of MF, dyskinesia, and FoG in an unsupervised scenario.

We conducted an observational, open-label study in our outpatient Movement Disorders Unit to assess this both issues of the sensor, under ambulatory conditions of use.

## Methods

### Study participants and inclusion criteria

Patients diagnosed with PD according to the UK PD Society Brain Bank criteria who routinely visited our unit, were recruited prospectively between September 2019 and December 2019 [48]. All participants met the following inclusion criteria: aged 30–80 years, were taking levodopa, had MC or a "suboptimal" motor state, and agreed to participate. A suboptimal motor state was considered when MF or WO were not clearly manifested according to the patient and the UPDRS-III motor scale of the patient was 3 points worse than the evaluation in the previous visit (the change considered by the literature as clinically significant). Participants were excluded if it was deemed by the interviewer's clinical judgment that they were unable to provide valid responses. Exclusion criteria also included unavailability to perform the study because of hospitalization or diagnosis of serious illness.

### Design, procedures, and clinical assessment

Socio-demographic and clinical data were collected. Standardized neurological assessment included the clinical interview, the Unified Parkinson's Disease Rating Scale (UPDRS)- II, UPDRS-III, UPDRS-IV, and the H&Y Staging of PD.

Patients were trained on how to use the device. The WIS was placed on their left hip in the first visit, and for one week, twelve hours a day (Fig 2).

After one week, the sensor was returned and WIS information was evaluated jointly by the neurologist and the patient. Treatment adjustments were indicated mainly according to best clinical practice guidelines but in some cases, device information reinforced these treatment decisions. The patients were analysed into two groups according to whether treatment changes had been indicated or not. In addition, the WIS satisfaction through QUEST scale was evaluated.

After 11 weeks, the WIS was again delivered for patients to use for a 7-day period, twelve hours per day. WIS information was shared with the patient. Assessment of FoG, Off-Time, and UPDRS II-IV and the SUS questionnaire were also completed (Fig 2).

### Patient's training on how to use the sensor

Patients were asked to place the WIS on their left hip during the day for 12 hours. They were encouraged to continue with their usual activities as much as possible. For a better understanding of the WIS information, patients were asked to record their ADL each hour. However, we asked them to take off de WIS if a long car journey was done, or they practiced physical activity. They were trained to push the unique button of the system when they took a levodopa dosage.

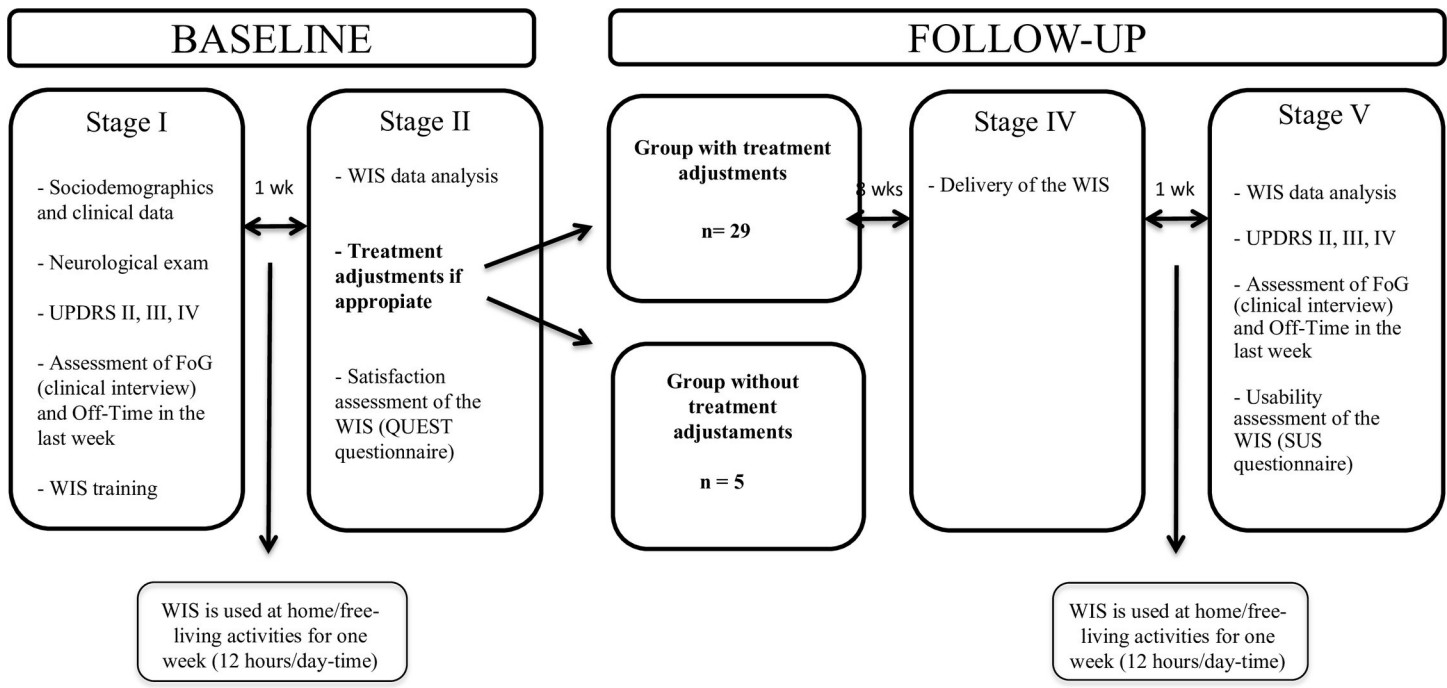

**Fig 2. Design, procedures, and clinical assessment.**

## Equipment

We used the STAT-ON$^{TM}$ system (https://www.statonholter.com/) as a continuous monitoring system through new technologies. The system is a Medical Device Class IIa (Council Directive 93/42/EEC of 14 June 1993) being only prescribed and provided from the clinician to the patient. The STAT-ON$^{TM}$ system consists of a monitoring device, its base charger, a belt, and a smartphone application (App). According to the Instructions for Use, The STAT-ON$^{TM}$ is a waist-worn inertial recorder, configured by a doctor and used by the patient for clinical, ambulatory, or home environments, that collects the results of the motor disorders and events of PD patients for a period. The smartphone can be property of the neurologist or the centre. The device continuously collects the inertial signals of the patient's movement, processes them in real-time by means of artificial intelligence algorithms, and stores the results in its internal memory. When the patient returns the sensor, the App connects to the STAT-ON$^{TM}$ device via Bluetooth and WIS data are downloaded and converted into structured report that can be emailed to any user or neurologist (Fig 1A–1D).

The App can only be managed by the health staff and can be installed on any smartphone or tablet (Android 5 or higher) or iOS for Apple devices (iOS 10.2 or higher). STAT-ON$^{TM}$ App is used both for configuring the system and for downloading the data previously generated by the sensor. The distance between the ground and the user's hip (leg length), the age and the H&Y Off stage are needed for a use configuration.

## Assessment of motor fluctuations, dyskinesia, FoG and patient-based Off-Time

Motor fluctuations were assessed by UPDRS-IV (item 39) while assessment of dyskinesia was based on item 35 of UPDRS-IV. The presence of FoG was evaluated according to the clinical interview.

## Data WIS analysis

**Selection of monitored days.** The analysis was performed selecting the 4 days with more hours in Off-Time. Although patients were asked to wear the sensor for 12 hours, the time of use during the day was variable. Therefore, all those days with less than 8 hours of monitoring were omitted. The variation of having days with 8 to more than 12 hours of monitoring was minimized using the percentage of time in OFF with respect to the total time monitored. Besides, in our experience there are some patients with fewer hours monitored that put on the sensor later in the morning. This can lead to missing data of the morning-Off's. The most appropriate and simple criterion to minimize this effect and not introduce bias in the sample was to compare the 4 worst days of each patient in terms of the percentage of OFF with respect to the monitored time.

**Analysis of sensor data for the correlation with clinical data.** The outcome rate of the sensor's information is provided every 30 minutes. This means that if 12 hours of monitoring are captured per day for 7 days, a total of 168 outcomes are obtained. Obviously, this is not directly comparable with a single clinical evaluation. In order to obtain a single diagnostic data for all the monitoring, different criteria and aggregations have been applied, which we detail below. In our experience, the most natural way to aggregate the sensor information is through a timeframe of one day. This means that the percentage of the different variables (Off, On, Intermediate- Time or dyskinesia) have only been computed if the monitoring period is more than 8h per day. From these daily measurements, we analyse the 4 worst days with higher OFF percentages. Finally, and with the aim of normalising the data amongst the patients, from these obtained 4 days, we calculate the mean of all the parameters. This way, it is obtained in a single data diagnosis which could be comparable to the clinical evaluation carried out in the consultation.

**Factor I of UPDRS-III.** UPDRS-III-Factor I is one of the parameters of the UPDRS that correlates better with the sensor signals. The UPDRS-III items clustered in the Factor-I are speech, facial expression, arising from a chair, gait, postural instability, and body bradykinesia [39,46,49]. The fluidity of the stride gait is a continuous variable, calculated for each group of strides of the patient, and which is the basis for calculating motor states [50–52]. In the present study we analysed the correlation of the Factor I and the mean fluidity of the stride. This analysis was evaluated in all the patients that completed the two monitorizations (after and before treatment changes).

## Statistical methods

Sensor variables were measured at baseline and three months later. The changes in these variables were compared between the patients in whom some therapeutic intervention was performed versus those that did not. Univariate comparisons were performed between both groups. The statistical tests included Chi-squared test, U-Mann-Whitney test, and t-student test as appropriate. Spearman correlation was applied for the analysis of Factor I of UPDRS-III. For the agreement analysis of MF, dyskinesias and FoG, the proportion of observed agreement and the Kappa coefficient was estimated for every pair of observers and every symptom. Statistical significance was set at $p < 0.05$. SPSS 24.0 statistical software (SPSS, Chicago, IL) was used for all statistical analyses.

## Results

### Characteristics of PD sample and WIS parameters at baseline

Thirty-nine PD patients were included in the study [mean (SD) age of 69 (8) years, 56% men, mean disease duration PD 8 (3.5) years]. All PD patients were treated with levodopa. Twenty-nine PD patients (74%) had predictable MF according to UPDRS-IV Q36, while 7.7% had

unpredictable MF. Seventeen patients (44%) had dyskinesia and twelve patients (31%) had FoG according to the clinical interview. Details regarding, demographics, motor state and therapies of sample, according the two groups (therapeutic intervention vs not) are shown in Table 1.

The WIS captured On, Off and intermediate state in the majority of patients (95–97%).

Characteristics regarding the motor state, the MC and patients' gait parameters according to the sensor are detailed in Table 2.

At the 3-month follow-up visit, five dropouts (13%) occurred: in three patients due to inter-current diseases and in two patients due to unavailability to perform the study, all of them not related to PD.

## Observed agreement of MF, dyskinesia and FoG at baseline

The system detected MF in thirty-seven patients (95%), whilst dyskinesia and FoG were detected in fifteen (41%), and nine (23%), respectively.

However, the kappa coefficient for MF, dyskinesia and FoG was 0.089, 0.318 and 0.481 respectively (Table 3).

## PD groups according to treatment adjustments

The treatment adjustments group included 29 patients was (85%) while there were 5 patients (15%) in the PD group without treatment adjustments. These 5 patients denied a modification

**Table 1. Characteristics of the study sample.**

|  | Change of treatment N = 34 | No change of treatment N = 5 | P Value |
|---|---|---|---|
| Age, years | 70 (6) | 63 (13) | 0.316 |
| Male, n (%) | 19 (56) | 3 (60) | 0.862 |
| Years from PD diagnosis | 7 (4.1) | 6 (2.3) | 0.554 |
| Years of PD evolution | 8 (3.7) | 7 (1.6) | 0.259 |
| H&Y |  |  | 0.427 |
| 1, n (%) | 2 (6) | 0 |  |
| 2, n (%) | 15 (44) | 1 (41) |  |
| 2.5, n (%) | 14 (41) | 4 (46) |  |
| 3, n (%) | 3 (9) | 0 |  |
| PD drug Treatment |  |  |  |
| L-DOPA use, n (%) | 34 (100) | 5(100) | - |
| DA agonist use, n (%) | 22 (65) | 3 (60) | 0.838 |
| MAOIs use, n (%) | 21 (62) | 4 (80) | 0.427 |
| COMT inhibitor use, n (%) | 13 (38) | 2 (40) | 0.940 |
| Amantadine, n (%) | 2 (5) | 0 | 0.578 |
| L-DOPA total daily dose | 576 (260) | 490 (134) | 0.475 |
| L-DOPA equivalence total daily dose | 705 (292) | 632 (343) | 0.614 |
| Advanced therapies, n (%) | 2 (5) | 0 | 0.578 |
| UPDRS-II on state | 9.8 (7) | 9.2 (4) | 0.848 |
| UPDRS-III on state | 21.6 (9.3) | 14.8 (9.6) | 0.194 |
| UPDRS-IV | 4.2 (2.3) | 3.9 (1.7) | 0.219 |

Values expressed as mean (standard deviation) except when indicated otherwise.

Abbreviations: H&Y = Hoehn and Yahr; PD = Parkinson disease; L-DOPA = Levodopa; DA = Dopaminergic; MAOI = Catechol-O-Methyltransferase Inhibitor:
UPDRS = Unified Parkinson's Disease Rating Scale.

**Table 2. Characteristics of the motor state, the motor complications, FoG, and gait parameters of the patients detected by the Wearable Inertial Sensor at baseline.**

| | Change of treatment N = 34 | No change of treatment N = 5 | P Value |
|---|---|---|---|
| Motor state | | | |
| Patients with Off state, n (%) | 33 (97) | 4 (80) | 0.106 |
| Mean percentage of time in Off state | 27 (10) | 17 (11) | 0.048* |
| Patients with On, n (%) | 32 (94) | 5 (100) | 0.578 |
| Mean percentage of time in On state | 27 (10) | 39 (14) | 0.026* |
| Patients with intermediate state, n (%) | 33 (97) | 5 (100) | 0.698 |
| Mean percentage of time in intermediate | 20 (6) | 17 (8) | 0.420 |
| Mean percentage of time without walking | 25 (10) | 26 (8) | 0.860 |
| Dyskinesia, n (%) | 16 (47) | 0 | 0.046* |
| Number of dyskinesias per day | 9 (11) | 2 (1) | 0.002* |
| FoG, n (%) | 8 (23) | 1 (20) | 0.861 |
| Number de FoG episodes per day | 7 (26) | 5 (7) | 0.866 |
| Gait parameters | | | |
| Minute walking per day | 73 (22) | 103 (21) | 0.008* |
| Steps per day | 8140 (2817) | 12320 (4046) | 0.006* |
| Cadence | 39 (2) | 39 (4) | 0.624 |
| Gait fluidity | 7 (1) | 9 (2) | 0.005* |

Values expressed mean (standard deviation) except when indicated otherwise.

Abbreviations: FoG = Freezing of gait.

Continuous variables were compared using t-Student test. Categorial variables were compared using the Chi-square test.

* p<0.05.

of the treatment scheduled. In fact, we usually face this scenario in clinical practice when dealing with PD patients without having sensor information. Sometimes the patient doesn't want to change the treatment because of their own decision because the intensity of Off or other PD symptoms are not so bothersome.

Levodopa was increased in 9 patients. In 2 patients the levodopa dose was decreased because adjustments of other dopaminergic drugs (opicapone and apomorphine infusion) were needed. Changes in levodopa schedule were indicated in 3 (10%) patients while a new treatment was added in 15 (52%) patients: opicapone 4, amantadine 3, oral/transdermic dopamine agonist 4, IMAOB (rasagiline/safinamide) 2, apomorphine infusion 1, antidepressant 1.

## Clinical and WIS data after treatment changes

In the group with changes in the treatment schedule (85%) the increase of the levodopa equivalent dose was 86.1 ± 69.7 mg while there was an improvement of the UPDRS-III On total score

**Table 3. Observed agreement between UPDRS-IV and wearable inertial sensor, regarding motor fluctuations, dyskinesia and FoG (at baseline).**

| | UPDRS-IV/clinical interview | WIS data | Observed agreement | Kappa coefficient |
|---|---|---|---|---|
| **Motor fluctuations, n (%)** | 29 (74) | 37 (95) | 74.4% | 0.089 |
| **Dyskinesias, n (%)** | 17 (44) | 16 (41) | 66.7% | 0.318 |
| **FoG, n (%)** | 12 (31) | 9 (23) | 76.9% | 0.481 |

Abbreviations: FoG = Freezing of gait; WIS = Wearable Inertial Sensor.

from 21.52 (+/- 9.82) to 16.69 (+/- 6.11). When comparing the mean percentages of Off-Time and On-Time between the two groups, there were statistically significant differences (Table 4). In the intervention group the percentage of Off-Time decreased from 28.30 ± 11.15 to 23.96 ± 13.37 while it increased in the group without treatment changes from 17.02 ± 11.64 to 23.07 (p = 0.007). Regarding the On-Time, it increased from 26.36 ± 10.83 to 29.22 ± 13.58 in the intervention group and decreased from 39.52 ± 14.56 to 29.82 ± 13.07 in the group without any intervention (p = 0.002). Differences were also observed between groups in terms of number of steps and GF with less decrease in the group in which treatment was changed (p = 0.008 and 0.004 respectively). Change in other sensor-based parameters (dyskinesia, FoG and minutes walking per day) was not significant.

When analysing the variables of On and Off sensor values in the "intervention group", it was found that the mean percentage of Off-Time among the patients who decreased their Off-Time (79% of patients) was -7.54 ± 5.26. The mean percentage of On-Time among the patients that increased their On-Time (59% of patients) was 8.9 ± 6.46.

When analysing the subgroup in which levodopa was increased (n = 9) against the group without treatment changes (n = 5), the WIS detected more On-Time in the patients in whom levodopa was increased (2.1 ± 6.7 vs– 9.7 ± 4.3; p = 0.003). More minutes walked (-1.1 ± 17.1 vs -22.0 ± 18.2; p = 0.039) and a greater number of steps (-281.3 ± 1976.8 vs– 3119.8 ± 1609.8; p = 0.014) were also found in this levodopa group. The GF worsen in the group without treatment changes (-0.8 ± 0.4) and remained unchanged in the levodopa group (0.0 ± 0.3) (p = 0.004).

## Discussion

New technologies and wearable sensors based on machine-learning approaches can help to capture the response to dopaminergic treatments and provide complementary information

**Table 4. Comparisons of the observed changes in WIS parameters between the two groups (treatment changes vs not) at baseline and after 3 months.**

| | Group with treatment changes n = 29 | | | Group without treatment changes n = 5 | | | P Value |
|---|---|---|---|---|---|---|---|
| | Basal | Month 3 | Change | Basal | Month 3 | Change | |
| **Clinical Outcomes** | | | | | | | |
| UPDRS-III ON | 21.52 (9.82) | 16.69 (6.11) | - 4.89 (6.88) | 14.80 (9.62) | 16.20 (7.79) | 1.4 (3.20) | 0.015* |
| Factor I (sub-scale UPDRS III) | 7.19 (3.9) | 5.94 (3.15) | -1.25 (1.63) | 6.0 (3.32) | 5.0 (3.53) | -1.0 (1) | 0.48 |
| UPDRS- II ON | 10.2 (8.1) | 9.3 (7.5) | -1.0 (3.0) | 9.2 (4.2) | 8.8 (4.7) | -0.4 (0.5) | 0.628 |
| UPDRS-IV | 4.1 (2.3) | 4.3 (2.1) | 0.2 (1.6) | 3.0 (1.7) | 2.2 (1.1) | 0.8 (0.8) | 0.080 |
| **WIS parameters** | | | | | | | |
| Mean % time in Off state | 28.30 (11.15) | 23.96 (13.37) | -4.34 (8.31) | 17.02 (11.64) | 23.07 (15.84) | 6.05 | 0.007* |
| Mean % time in On state | 26.36 (10.83) | 29.22 (13.58) | 2.86 (9.44) | 39.52 (14.56) | 29.82 (13.07) | -9.70 (4.32) | 0.002* |
| Mean % time in Intermediate state | 20.0 (7.3) | 18.7 (7.1) | 1.2 (6.8) | 17.3 (8.3) | 24.7 (6.7) | 7.4 (11.0) | 0.103 |
| Mean % time in inactivity state | 25.4 (11.2) | 28.1 (12.9) | 2.7 (6.6) | 26.2 (8.5) | 22.4 (6.8) | -3.7 (8.5) | 0.126 |
| Dyskinesia per day | 9.27 (9.74) | 13.17 (13.71) | 3.90 (9.34) | 2.27 (1.66) | 1.95 (2.74) | -0.32 | 0.206 |
| FoG per day | 8.52 (28.35) | 4.81 (13.26) | -3.71 (15.52) | 5.46 (7.12) | 5.57 (8.81) | 0.11 | 0.610 |
| Minute walking per day | 75.64(22.90) | 71.97 (24.65) | -3.67 (18.69) | 103.26 (21.41) | 81.26 (23.97) | -22 | 0.061 |
| Steps per day | 8396(2890) | 7840 (2878) | -556 (1871) | 12320 (4046) | 9200 (3646) | -3120 | 0.008* |
| Gait fluidity | 7.37 (1.35) | 7.43 (1.54) | 0.06 (0.61) | 9.38 (2.03) | 8.57 (1.93) | -0.81 | 0.004* |

Values expressed as mean (standard deviation).

Comparisons of the observed changes in WIS parameters between groups were made using Mann-Whitney U test *p<0.05.

besides the clinical scales and patient's subjective opinion about the treatment response. In the present study, the STAT-ON$^{TM}$ system can detect significant sensor-based changes in terms of a decrease in the percentage of Off-Time, and increase of On-Time, number steps and GF after three months of having modified the treatment schedule. Besides, we have obtained a value of $8.9 \pm 6.46$ for the percentage of On-Time and $-7.54 \pm 5.26$ for the Off-Time percentage. These preliminary results will need further replication for future studies with this sensor when analysing the magnitude of effect of interventions (either pharmacological or non-pharmacological such as physiotherapy). We have also found that the number of steps increased significantly, yet the minutes walking per day did not, meaning that the walking speed increased. Hence, with the wearable sensors not only the Off-Time/On-Time is important to analyse, but also gait variables such as GF and gait speed.

Other studies with different wearable-based technology have tried to demonstrate data-sensor changes after dopaminergic treatments in real-world settings [20,21,24–26,28–30]. Wrist sensors can overestimate symptoms more than waist-worn sensors, particularly tremor and dyskinesia [53]. Nonetheless, if the objective is to monitor the change measured by the sensor rather than accurately record PD symptoms, this can be achieved with either wrist or waist sensors [53]. In a recent study were PD patients used a smartphone App while doing 5 activities (voice, finger tapping, gait, balance, and reaction time), a mobile PD score (mPDS) derived from a novel machine-learning approach, captured improvement in response to dopaminergic therapy [20]. The watch device Personal KinetiGraph (PKG) has been widely used in the setting of routine clinical practice to identify a clinically significant response to levodopa [21], assess the changes in the dopaminergic oral therapy [24,28,29], identify candidates for second line therapies [24,50], or determine the changes in motor fluctuations following deep brain stimulation [26].

Wearable technologies provide additional and useful objective data that can enhance an enrichment of the clinical decision-making process, empowering the patient in the treatment decisions [22,23]. In our study, the treatment decisions were done according to the best clinical practice guidelines, but we are aware that the STAT-ON$^{TM}$ information could have influenced these treatment decisions. However, in the present study the WIS report was available for all the patients in the second visit. Hence, we don't have measured how the impact of the STAT-ON$^{TM}$ information can influence in the treatment decisions. Nonetheless, in the ongoing multicentre, randomised clinical trial MoMoPa-EC, in which the STAT-ON$^{TM}$ is used, this issue will be analysed because there are three study arms (sensor information vs Hauser diaries vs clinical information only collected in the visit) [35]. In fact, other studies using wearable technology demonstrate that more treatment plans are performed with the additional information captured by the sensors, probably improving the patient dialogue [24,25,28,29].

This device is only able to detect the Off-Time by analysing the bradykinetic gait while the patient is walking and wearing the sensor [36,37]. According to the previous studies of this sensor, the sensitivity and specificity of the On/Off STAT-ON$^{TM}$ algorithms are of 97% and 88% respectively [39,40,42]. According to our results, this wearable system detects Off state in almost all the patients (95%) at baseline. Still, there was no agreement between the clinical and the WIS results when applying the Kappa analysis. Nonetheless, these results could be influenced by the fact that we don't have used the Hauser diaries for the correlation. In this line, in the previous studies for the elaboration of the STAT-ON sensor algorithms, patients that didn't experience MF or were unable to recognize them, were excluded of the studies [42,47]. On the contrary, in the present pilot study we have included patients with initial MF or that they did not manifested WO. However, a recent sub-analysis of the MoMoPa-EC study that analyse the agreement between the Hauser diaries and the STAT-ON sensor, show an

intraclass correlation coefficient (ICC) of both methods of 0.57 (95% CI: 0.3–0.73) for the Off-Time and 0.48 (95% CI: 0.17–0.68) for the On-Time [54].

The disagreement in the detection of MF between the clinical scale and the sensor, especially in the scenario of the daily-clinical practice, opens the discussion about whether the sensor is detecting the problem before the patient understands the reason for it. The transition from the best patient's motor state to worst motor state can be gradual and ambiguous for some patients, leading to imprecise evaluation [4]. Besides, some barriers exist in the clinical practice when evaluating the off periods. These barriers could be lack of awareness of the symptom among patients, cognitive impairment, reluctance to discuss symptoms, caregiver absence, lack of time's physician to evaluate all the symptoms or physician's lack of appreciation of the Off periods [55]. In this regard, the use of wearable medical devices can be crucial and potentially help the clinicians in the detection of MF [56]. Future studies with larger samples using specific WO scales and exploring the ability of STAT-ON$^{TM}$ of capturing initial MF are mandatory.

With regard to dyskinesia, the sensitivity and specificity for strong or mild trunk dyskinesia are 95% and 93% respectively, and the sensitivity is lower (39%) for mild limb dyskinesias [44]. In the sub-analysis of the MoMoPa trial mentioned before, the ICC of the Hauser diaries and the STAT-ON was 0.48 (95% CI: 0.17–0.68) [54]. The STAT-ON cannot detect all the type of dyskinesia along all the time of monitoring. Besides, the sensor only detects dyskinesia when the patient is not walking. This means, when the bradykinesia algorithm is not working. The kappa agreement for dyskinesia in the present study is of 0.318. Disagreement in dyskinesia can be explained because either the patient may not be aware of their dyskinesia, or the patient is doing an activity that might imitate a "false dyskinesia" like dancing. In the present study we tried to control the patient's activities in a simple diary. However, we couldn't control at all the activities potentially responsible for "false positive dyskinesia" such as cleaning, sweeping or going by car or public transport. With respect to the FoG, in the previous algorithm validation study of this WIS, the sensitivity and specificity were around 85% [43]. To our knowledge, there are still no studies that have analysed the agreement between the clinical detection of FoG and the STAT-ON$^{TM}$ sensor detection of FoG in an unsupervised scenario. Although in our study there is the limitation that specific FoG detection clinical scales have not been used, a kappa coefficient of 0.481 have been obtained. Hence, it will be of interest to have future results analysing this issue.

The results of the Spearman correlation regarding the UPDRS-III-Factor I and the STAT-ON$^{TM}$ measures have been previously studied. In the study of Rodríguez-Molinero et al, the correlation between the algorithm outputs and the UPDRS-III- Factor I was -0.67 (p < 0.01) [46]. In this study the comparison was directly between the Factor I and the GF while in our study we have obtained the variation of the mean GF of the same patient in the two monitorizations, which has 0.6. Although these results cannot be directly compared, the correlation obtained in our study shows consistent results with the previous ones. Recently, Pérez-López and collaborators, have found a Spearman correlation of– 0.63 (p> 0.001) between the mean fluidity and the UPDRS-III- Factor I [54].

The present work has several limitations. Due to the design of the study, heterogeneity in both the PD sample and the treatment groups couldn't have been avoided. Moreover, we haven't used other specific clinical scales for evaluating the WO, dyskinesia or FoG, nor the Hauser diaries have been used. Finally, the interesting point of the study that is performed in a "free-living-environment" implies that some false positives are unavoidable.

Despite the limitations of our study, we think that this wearable can provide additional value to PD neurologists for a better understanding of the PD patient 's motor state and a guiding for treatment decisions in the setting of routine clinical practice [8,23,57,58].

In summary, it's feasible to use this sensor for monitoring PD treatment under ambulatory conditions. This system could serve as a complementary tool to assess PD motor symptoms and motor complications, although more studies with larger and homogeneous PD samples should be completed.

## Acknowledgments

We thank all the patients and their caregivers for participating in this study.

## Author Contributions

**Conceptualization:** Nuria Caballol, Àngels Bayés.

**Data curation:** Nuria Caballol, Anna Prats, Montserrat Martín-Baranera.

**Formal analysis:** Anna Prats, Montserrat Martín-Baranera.

**Investigation:** Nuria Caballol, Àngels Bayés, Paola Quispe.

**Methodology:** Nuria Caballol, Àngels Bayés, Anna Prats.

**Project administration:** Nuria Caballol.

**Resources:** Paola Quispe.

**Supervision:** Nuria Caballol, Àngels Bayés, Montserrat Martín-Baranera.

**Validation:** Nuria Caballol, Àngels Bayés.

**Visualization:** Nuria Caballol.

**Writing – original draft:** Nuria Caballol, Àngels Bayés.

**Writing – review & editing:** Nuria Caballol, Àngels Bayés.

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
