## [Decision Letter · Decision Letter 0]

25 Jul 2022

PONE-D-22-12897Feasibility of a wearable inertial sensor to assess motor complications and treatment in Parkinson’s diseasePLOS ONE

Dear Dr. Caballol,

Thank you for submitting your manuscript to PLOS ONE. After careful consideration, we feel that it has merit but does not fully meet PLOS ONE’s publication criteria as it currently stands. Therefore, we invite you to submit a revised version of the manuscript that addresses the points raised during the review process.

For Lab, Study and Registered Report Prot

We look forward to receiving your revised manuscript.

Kind regards,

Keisuke Suzuki, MD, PhD

Academic Editor

PLOS ONE

Journal Requirements:

2. PLOS requires an ORCID iD for the corresponding author in Editorial Manager on papers submitted after December 6th, 2016. Please ensure that you have an ORCID iD and that it is validated in Editorial Manager. To do this, go to ‘Update my Information’ (in the upper left-hand corner of the main menu), and click on the Fetch/Validate link next to the ORCID field. This will take you to the ORCID site and allow you to create a new iD or authenticate a pre-existing iD in Editorial Manager. Please see the following video for instructions on linking an ORCID iD to your Editorial Manager account: https://www.youtube.com/watch?v=_xcclfuvtxQ.

3. Please upload a new copy of Figure 1 as the detail is not clear. Please follow the link for more information: https://blogs.plos.org/plos/2019/06/looking-good-tips-for-creating-your-plos-figures-graphics/" https://blogs.plos.org/plos/2019/06/looking-good-tips-for-creating-your-plos-figures-graphics/

Reviewers' comments:

Reviewer's Responses to Questions

**Comments to the Author**

1. Is the manuscript technically sound, and do the data support the conclusions?

Reviewer #1: Partly

Reviewer #2: Partly

2. Has the statistical analysis been performed appropriately and rigorously? 

Reviewer #1: I Don't Know

Reviewer #2: Yes

3. Have the authors made all data underlying the findings in their manuscript fully available?

Reviewer #1: No

Reviewer #2: Yes

4. Is the manuscript presented in an intelligible fashion and written in standard English?

Reviewer #1: No

Reviewer #2: No

5. Review Comments to the Author

Reviewer #1: The reporting of real life use of inertial sensor data for monitoring motor symptoms in Parkinson's disease is much needed and important for validation of their use in clinical routine. This manuscript adds relevant information from a pragmatic use of the STAT-ON wearable device. The focus of the authors is on evaluating if the measuring system is responsive to treatment changes and they report significant improvements in several measures in a group of patients (n=34) where a change of treatment was indicated compared to a small group where a change was not indicated (n=5). Patients were systematically assessed at baseline as well as three months after a decision to change or not change treatment, regarding off-time and FOG (clinical interview and STATON), UPDRS II-IV and STATON measures.

1. Although there are several interesting observations, it appears to me that some data of interest are not reported or are "hidden". There were changes in STATON time in ON and OFF-state, but were these detected in the clinical interviews and/or UPDRS IV? The description of UPDRS data (p11) at 3Mo is unclear and as a reader I am not always sure if off-time refers to UPDRS or WIS. This is particularly important as authors discuss the possibility of over-detection using sensors. Please make clearer distinction between WIS-results and clinical and rating scale outcomes at 3 Mo.

2. Other information that I find missing is the implemented changes in levodopa equivalent doses in the intervention group. The statement that levodopa was increased in 9 and lowered in 2 is not informative unless the resulting levodopa equivalent doses are given, although I suspect that it may actually refer to levodopa equivalent doses. Correct? Please report actual changes too.

3. I found it bewildering that increases and decreases in treatment intensity were both treated as "intervention". Because the direction of change in STATON measures is expected to be opposite with these two different interventions I would expect them to be reported separately. There is a mentioning of the levodopa increase group and a comparison with no treatment on page 12-13, but the statements there are not supported by statistical tests. Please consider to report interventions due to undertreatment and due to excessive effect (dyskinesia) separately.

4. Another analysis that I would have liked to see is a Cohen alpha for the agreement between the presence of FOG, motor fluctuations and dyskinesia as assessed with interview/UPDRS and STATON. Currently the proportions are reported (and are similar), but not the agreement between assessments.

5. It is somewhat unclear why some patients were not considered suboptimal enough to change treatment as the inclusion criteria for the study were that they had motor complications or suboptimal treatment effect. Please consider to clarify this since it makes it difficult to understand the "no interventions" group.

6. Authors state that treatment adjustments were made mainly according to best clinical practice. If the sensors do not add information that influences treatment choices their use can be questioned or at best have to be motivated in some other way. It would have been good to know in how many measurements information that was not available at the interview stage was added by the STATON device and if at any time this lead to changes or questioning of intervention strategy. I realize that the study should have had a stage where a preliminary treatment strategy was established before the STATON measurement to formally address that, but maybe something could be said, or at least commented on in the discussion.

7. Please clarify the selection of WIS data (p7 L14). 4 days is not the regularly reported data in STATON reports, and what does "more OFF" mean. Why was this selection of a subset of data made?

8. Sensor based percent OFF-time and ON-time is reported, presumably as percent of monitored time. This may need some explanation. Can for example a 4.5% increase in ON-time be translated to 4.5% of 12h (i.e. approx 0.5h)?. The time in Intermediate state (grey) is only reported at baseline, why not report change in intermediate time?

9. P7 L22 - A number or primary outcome measures are listed, but no secondary. If all these measures are primary, was adjustment for multiple testing done? I suggest presenting data as exploratory and without multiple adjustment instead.

Some more editorial and linguistic concerns:

1. The objective (abstract) is stated as "To analyse the ability of a wearable inertial sensor (WIS) to detect changes in the

On/Off-Time, dyskinesia, freezing of gait (FoG) and gait parameters after treatment" This objective is not met in the manuscript. It is the responsivity of WIS measures to treatment change that is studied. You would have had to compare clinical and WIS outcomes at three months to say something about this objective.

2. the word "tracked" is used in the sense "detected" (for example P2 L15) and not with its usual meaning (to follow or pursue in time or space). Consider revising.

3. MF is not explained at first occurence.

4. Introduction L7 (p3) - either define early in disease or omit (not sure it matters when fluctuations occur - you still need to recognize them).

5. p3 L14 - Fulfilling

6. p3 L24 - it is a defensive revolution it wearables will only change the way we monitor disease and not manage disease.

7. P4 L21-22 - "ability of the system to track sensor-based changes after treatment adjustments." Unclear, consider revising.

8. p5 L14 - does this mean subjects had no comorbidity at all?

9. p8 L3 (and onwards) I doubt Factor I of UPDRS III is an established measure. I for one, would prefer that you explained which items are included in this subset.

10. p8 L11 - any reasons for not reporting unpredictable MF?

11. p11 L16: I believe ", and" should be replaced with ", an".

12. p11 Maybe consider presenting mean ON-time and OFF-time before proportion of patients who increased or decreased their time in the respective state. (See also major concern no 1).

13. p11 L22-23. This sentence is unclear. You state increase in one group and decrease in the other, but there is only one p-value, so presumably you mean that there is a difference between the groups. (Unless you made separate analyses in each group over time).

14. p13 L3 - replace correspondence with correlation

15. Consider using "measure" or "variable" instead of parameter in multiple places in the Ms.

16. p14 L15 - The patients in the Kinesia 360 study were randomized so there was no difference in indications for Kinesia monitoring

17. p14 L22 - regarding detection of problems before the patient understands them, I think I agree. However, the way it is worded now one can question if there is a reason to detect things that are unrecognized since patients are treated symptomatically. Maybe underline that it makes it possible to correctly identify a problem before the patient understands the reason for it. It is well known that OFF is a phenomenon that is difficult to communicate and that problem is a barrier between patient and physician (e.g. https://doi.org/10.1371/journal.pone.0215384).

18. p15 L7 - Is there a difference between 0.6 and 0.67? If so, how does the different circumstances of your study and previous explain that? I do not understand.

19. p15 L10 - I don't think you have explained the limitations of the study...

20. Finally I will be cheeky enough to ask what the magic feature of machine learning is that warrants the special mentioning on p15 L11? If some other method works, would it be inferior because it is not machine learning? PKG is not based on machine learning and I think there is a reasonable body of evidence to suggest that it can still provide additional value to neurologists in some situations.

Reviewer #2: Summary

Caballol et al. present an observational study of 29 PD patients who wore a STAT-ON wearable inertial sensor for one week before treatment and one week three months after treatment and 5 PD patients whose treatment did not need adjusting. From the one week of observation, the four days with most time spent in Off were selected for further analysis. Statistically significant changes were seen in the percentage of Off-Time, number of steps and gait fluidity in the group of patients with intervention. The aim of this paper was to show that wearable sensors can be used as a complementary tool to monitor therapy impact and assess PD motor complications. The following aspects may help improve the quality of the manuscript:

General Comments

The authors should thoroughly revise the language and make sure sentences end with full stops to improve readability and comprehension. I have only mentioned some from the first few paragraphs in the minor comments.

If the number of steps increased significantly, yet the minutes walking per day did not, this means that the walking speed increased. This could be an interesting aspect for the discussion.

Major

· Introduction:

o Could you emphasise what the challenges wearable sensor-based systems have and why you believe there is “lack of external validation”?

o Why did the authors focus only on the STAT-ON sensor? More information about wearables the advantages and disadvantages of the data that can be gathered would be useful.

· Methods:

o Why did the authors select only 4 days with more hours in off-time? Does this not introduce a bias by selecting the days with more off-time hours, especially if the percentage of off-time is compared before and after intervention and is one of the primary outcomes?

o Given different group sizes, or, rather, difference in variance between groups, the Welch test should be used instead of the t-test.

o How are variables such as dyskinesia per day, time in on/off, FoG per day calculated? Is there maybe a paper that could be cited where the algorithms from STAT-ON are explained?

o Since multiple tests were executed, a correction for multiple testing (e.g. Bonferroni) should be employed.

· Results:

o Pg. 13, lines 3-7: in order to correlate sensor parameters and clinical evaluation parameters, either the mean (or other summary statistic) of the repeated measure (sensor parameter) or repeated measure correlation must be used to test for correlation between the two parameters, otherwise the significance is inflated. Were repeated measures or means per patient for the whole timeframe used? It is unclear from the text.

· Data Availability: the authors state that “all relevant data are within the manuscript and its Supporting Information files”. Does this mean the sensor data will be uploaded in the supporting information files? I would highly recommend to make the data publicly available by means of a public repository.

Minor

· Authors state that “PD patients wore a validated WIS for one week. WIS data were analysed before treatment and three months after therapeutic changes.” (Page 2, line 11). Rephrasing would help, as it currently gives the impression that the WIS were worn for one week total. E.g.: “… WIS for one week before treatment and one week three months after therapeutic changes.”

· Please take a look at your figure numbering, in the manuscript you mention Figure 1 c-d, yet they are not present among your figures.

· Table 1: please refrain from expressing percent in parentheses if standard deviation is in parenthesis for other variables.

· Page 2, line 14 missing closing parenthesises and full stop. Line 15: unclear what MF is, as used first time.

· Page 3, line 14: Not sure if “Full-filling” is the right formulation, “Completing” could be better.

6. PLOS authors have the option to publish the peer review history of their article (what does this mean?). If published, this will include your full peer review and any attached files.

Reviewer #1: No

Reviewer #2: No

---

## [Author Response · Author response to Decision Letter 0]

14 Nov 2022

Response to Reviewers' comments:

Reviewer #1: 

The reporting of real-life use of inertial sensor data for monitoring motor symptoms in Parkinson's disease is much needed and important for validation of their use in clinical routine. This manuscript adds relevant information from a pragmatic use of the STAT-ON wearable device. The focus of the authors is on evaluating if the measuring system is responsive to treatment changes and they report significant improvements in several measures in a group of patients (n=34) where a change of treatment was indicated compared to a small group where a change was not indicated (n=5). Patients were systematically assessed at baseline as well as three months after a decision to change or not change treatment, regarding off-time and FOG (clinical interview and STATON), UPDRS II-IV and STATON measures.

1. Although there are several interesting observations, it appears to me that some data of interest are not reported or are "hidden". There were changes in STATON time in ON and OFF-state, but were these detected in the clinical interviews and/or UPDRS IV? The description of UPDRS data (p11) at 3Mo is unclear and as a reader I am not always sure if off-time refers to UPDRS or WIS. This is particularly important as authors discuss the possibility of over-detection using sensors. Please make clearer distinction between WIS-results and clinical and rating scale outcomes at 3 Mo.

We thank the reviewer these important comments. We have added in the table 3, more clinical outcomes (UPDRS II, IV) that were lacking. We have also made a clearer distinction between clinical vs WIS results in the table (see table 4, page 15). 

Since we don’t have done the correlations between the changes of WIS parameters after treatment changes and the clinical interview/UPDRS IV, we have redefined our objectives, according to the comments below. 

2. Other information that I find missing is the implemented changes in levodopa equivalent doses in the intervention group. The statement that levodopa was increased in 9 and lowered in 2 is not informative unless the resulting levodopa equivalent doses are given, although I suspect that it may actually refer to levodopa equivalent doses. Correct? Please report actual changes too.

We appreciate the reviewer’s comment. We report the levodopa equivalent doses in the intervention group. The increase in the levodopa equivalent dose was 86.1 � 69.7 mg (page 14, line 6). 

3. I found it bewildering that increases and decreases in treatment intensity were both treated as "intervention". Because the direction of change in STATON measures is expected to be opposite with these two different interventions I would expect them to be reported separately. There is a mentioning of the levodopa increase group and a comparison with no treatment on page 12-13, but the statements there are not supported by statistical tests. Please consider reporting interventions due to undertreatment and due to excessive effect (dyskinesia) separately.

Thank you for pointing this out. In the “intervention group” there was an increase of the levodopa equivalent dose of 86.1 � 69.7 mg of levodopa equivalent dose and in the control group there was no increase at all. In 2 patients the levodopa dose was decreased because adjustments of other dopaminergic drugs (opicapone and apomorphine infusion) were needed. For this reason, these 2 patients were included in the “intervention group”. We have clarified this issue in the text (Page 13, line 19).

Sorry for the missing statistical tests of the levodopa subgroup. We have added all the values in the text. When analysing the subgroup in which levodopa was increased (n=9) against the group without treatment changes (n=5), the WIS detected more On-Time in the patients in whom levodopa was increased (2.1 � 6.7 vs – 9.7 � 4.3; p= 0.003). More minutes walked (-1.1 � 17.1 vs -22.0 � 18.2; p= 0.039) and a greater number of steps 

(-281.3 � 1976.8 vs – 3119.8 � 1609.8; p= 0.014) were also in this levodopa group.

The GF worsen in the group without treatment changes (-0.8 � 0.4) and remained unchanged in the levodopa group (0.0 � 0.3) (p=0.004). (Page 16, lines 1-7).

4. Another analysis that I would have liked to see is a Cohen alpha for the agreement between the presence of FOG, motor fluctuations and dyskinesia as assessed with interview/UPDRS and STATON. Currently the proportions are reported (and are similar), but not the agreement between assessments.

We appreciate the reviewer this important observation. We present the results of the agreement analysis between the presence of the motor fluctuations, dyskinesia and FOG according to the clinical interview based on the UPDRS-IV items and the WIS results (analysis with the Cohen kappa). See page 13, table 3. This analysis has been done only at baseline. We have discussed the agreement of MF, dyskinesia and FoG, along all the discussion. 

5. It is somewhat unclear why some patients were not considered suboptimal enough to change treatment as the inclusion criteria for the study were that they had motor complications or suboptimal treatment effect. Please consider clarifying this since it makes it difficult to understand the "no interventions" group.

Thank you for this comment. We have reviewed all the reasons for not doing treatment changes in the 5 patients without intervention. In all the cases the patients denied a modification of the treatment scheduled. In fact, we usually face this scenario in clinical practice when dealing with PD patients without having sensor information. Sometimes the patient doesn’t want to change the treatment because of their own decision because the intensity of Off or other PD symptoms are not so bothersome. We have added this information in the text. See page 13, lines 13-17. 

6. Authors state that treatment adjustments were made mainly according to best clinical practice. If the sensors do not add information that influences treatment choices their use can be questioned or at best have to be motivated in some other way. It would have been good to know in how many measurements information that was not available at the interview stage was added by the STATON device and if at any time this lead to changes or questioning of intervention strategy. I realize that the study should have had a stage where a preliminary treatment strategy was established before the STATON measurement to formally address that, but maybe something could be said, or at least commented on in the discussion.

Thank you for this pointing this out. Unfortunately, this issue has not been analyzed in our study. However, we have clarified this point in the discussion: “In our study, the treatment decisions were done according to the best clinical practice guidelines, but we are aware that the STAT-ONTM information could have influenced these treatment decisions. However, in the present study the WIS report was available for all the patients in the second visit. Hence, we don’t have measured how the impact of the STAT-ON information can influence in the treatment decisions. Nonetheless, in the ongoing multicentre, randomised clinical trial MoMoPa-EC, in which the STAT-ON is used, this issue will be analysed because there are three study arms (sensor information vs Hauser diaries vs clinical information only collected in the visit) [35]”. See page 17, lines 11-20. 

7. Please clarify the selection of WIS data (p7 L14). 4 days is not the regularly reported data in STATON reports, and what does "more OFF" mean. Why was this selection of a subset of data made

We thank the reviewer this comment that needs to be clarified. 

As the reviewer rightly point out, this is an aspect that has caused some discussion during the analysis of the data. We have clarified it as follows: “Although patients were asked to wear the sensor for 12 hours, the time of use during the day was variable. Therefore, all those days with less than 8 hours of monitoring were omitted. The variation of having days with 8 to more than 12 hours of monitoring was minimized using the percentage of time in OFF with respect to the total time monitored. Besides, in our experience there are some patients with fewer hours monitored that put on the sensor later in the morning. This can lead to missing data of the morning-Off’s. The most appropriate and simple criterion to minimize this effect and not introduce bias in the sample was to compare the 4 worst days of each patient in terms of the percentage of OFF with respect to the monitored time.”. See page 8, lines 13-22. 

8. Sensor based percent OFF-time and ON-time is reported, presumably as percent of monitored time. This may need some explanation. Can for example a 4.5% increase in ON-time be translated to 4.5% of 12h (i.e. approx 0.5h)?. The time in Intermediate state (grey) is only reported at baseline, why not report change in intermediate time? 

Thanks for pointing this out. Exactly, the percentage of On-Time is regarding the total time monitored. The main goal of this parameter is to provide the quantity of a motor state without considering the total time. We have modified the Figure 1 and we think that this issue can be easier to understand (see part “C” of Figure 1). 

We have added in table 4 the intermediate state (yellow in the STAT-ON report, see Figure 1) and the time with inactivity (grey in the STAT-ON report). See page 15. 

9. P7 L22 - A number or primary outcome measures are listed, but no secondary. If all these measures are primary, was adjustment for multiple testing done? I suggest presenting data as exploratory and without multiple adjustment instead.

We appreciate this comment. Because of the nature of our study (pilot study) our results are exploratory. According to this, we have modified the paragraph of Statistical Tests: 

“Sensor variables were measured at baseline and three months later. The changes in these variables were compared between the patients in whom some therapeutic intervention was performed versus those that did not. Univariate comparisons were performed between both groups. The statistical tests included Chi-squared test, U-Mann-Whitney test, and t-student test as appropriate. Spearman correlation was applied for the analysis of Factor I of UPDRS-III. For the agreement analysis of MF, dyskinesias and FoG, the proportion of observed agreement and the Kappa coefficient was estimated for every pair of observers and every symptom. See page 10, lines 1-9.

Some more editorial and linguistic concerns:

1. The objective (abstract) is stated as "To analyse the ability of a wearable inertial sensor (WIS) to detect changes in the On/Off-Time, dyskinesia, freezing of gait (FoG) and gait parameters after treatment" This objective is not met in the manuscript. It is the responsivity of WIS measures to treatment change that is studied. You would have had to compare clinical and WIS outcomes at three months to say something about this objective.

We thank the reviewer this important comment. We have now redefined better our objectives which are as follows: “To analyse the responsivity of the sensor measures (On/Off-Time, dyskinesia, FoG and gait parameters) after treatment adjustments. We also aim to study the ability of the sensor in the detection of MF, dyskinesia, FoG and the percentage of Off-Time, under ambulatory conditions of use”. Hence, we have modified the abstract and the introduction. 

2. the word "tracked" is used in the sense "detected" (for example P2 L15) and not with its usual meaning (to follow or pursue in time or space). Consider revising.

Thank you for your comment. We have avoided the word “tracked” in the text. 

3. MF is not explained at first occurrence.

Thank you for pointing this out. We have corrected it. 

4. Introduction L7 (p3) - either define early in disease or omit (not sure it matters when fluctuations occur - you still need to recognize them)

We agree with your comment. We have avoided this comment because it is confusing. 

5. p3 L14 – Fulfilling

Thank you. Corrected. 

6. p3 L24 - it is a defensive revolution it wearables will only change the way we monitor disease and not manage disease.

Thank you for this observation. We agree that it is speculative and is better to avoid this issue. 

7. P4 L21-22 - "ability of the system to track sensor-based changes after treatment adjustments." Unclear, consider revising.

Thank you. We have redefined the objective and changes the sentence as follows: “However, there has been no published data regarding the responsivity of the sensor's measures after treatment adjustments”. 

8. p5 L14 - does this mean subjects had no comorbidity at all?

Thank you for pointing this issue that needs to be clarified in the text. The patients had comorbidities, but we don’t describe them in the study. They were excluded if they were unable to perform the study because hospitalization or diagnosis of serious illness (see Page 6, lines 7-8). 

9. p8 L3 (and onwards) I doubt Factor I of UPDRS III is an established measure. I for one, would prefer that you explained which items are included in this subset.

Thank you for the comments regarding this important issue that needs a better explanation in the text. We have explained better which items are clustered in the Factor-I (speech, facial expression, arising from a chair, gait, postural instability, and body bradykinesia). See, methods Page 9, lines 16-18. 

10. p8 L11 - any reasons for not reporting unpredictable MF?

Thank you for your observation. We have added this information in the text (Page 10, lines 16). A 7.7% of the sample had unpredictable MF. 

11. p11 L16: I believe ", and" should be replaced with ", an".

Thank you. It has been corrected. 

12. p11 Maybe consider presenting mean ON-time and OFF-time before proportion of patients who increased or decreased their time in the respective state. (See also major concern no 1).

We thank the reviewer this comment. We have completed the results with this information which are part of the analysis of the “responsivity of the sensor”. “When analysing the variables of On and Off sensor values in the “intervention group”, it was found that the mean percentage of Off-Time among the patients who decreased their Off-Time (79% of patients) was -7.54 � 5.26. The mean percentage of On-Time among the patients that increased their On-Time (59% of patients) was 8.9 � 6.46” See page 14, lines 18-21.

13. p11 L22-23. This sentence is unclear. You state increase in one group and decrease in the other, but there is only one p-value, so presumably you mean that there is a difference between the groups. (Unless you made separate analyses in each group over time).

Thank you for your comment. We have made separate analysis in each group for Off-Time and On-Time over time and then we have compared the results. We have clarified this issue in the text as follows: “When comparing the mean percentages of Off-Time and On-Time between the two groups, there were statistically significant differences. In the intervention group the percentage of Off-Time decreased from 28.30 � 11.15 to 23.96 � 13.37 while it increased in the group without treatment changes from 17.02 � 11.64 to 23.07 (p= 0.007). Regarding the On-Time, it increased from 26.36 � 10.83 to 29.22 � 13.58 in the intervention group and decreased from 39.52 � 14.56 to 29.82 � 13.07 in the group without any intervention (p= 0.002).” (See page 14, lines 7-13). 

14. p13 L3 - replace correspondence with correlation

Thank you. It has been replaced. 

15. Consider using "measure" or "variable" instead of parameter in multiple places in the Ms.

Thank you for your suggestion. We have corrected it in the text. 

16. p14 L15 - The patients in the Kinesia 360 study were randomized so there was no difference in indications for Kinesia monitoring.

Thank you for pointing this out. We have omitted this conclusion as it was incorrect. 

17. p14 L22 - regarding detection of problems before the patient understands them, I think I agree. However, the way it is worded now one can question if there is a reason to detect things that are unrecognized since patients are treated symptomatically. Maybe underline that it makes it possible to correctly identify a problem before the patient understands the reason for it. It is well known that OFF is a phenomenon that is difficult to communicate and that problem is a barrier between patient and physician (e.g. https://doi.org/10.1371/journal.pone.0215384).

Thank you very much for this observation. We have clarified better this issue in the text as follows: “The disagreement in the detection of MF between the clinical scale and the sensor, especially in the scenario of the daily-clinical practice, this opens the discussion about whether the sensor is detecting the problem before the patient understands the reason for it. The transition from the best patient’s motor state to worst motor state can be gradual and ambiguous for some patients, leading to imprecise evaluation [4]. Besides, some barriers exist in the clinical practice when evaluating the off periods. These barriers could be lack of awareness of the symptom among patients, cognitive impairment, reluctance to discuss symptoms, caregiver absence, lack of time’s physician to evaluate all the symptoms or physician’s lack of appreciation of the Off periods [55]. Page 18, lines 13-21.

18. p15 L7 - Is there a difference between 0.6 and 0.67? If so, how does the different circumstances of your study and previous explain that? I do not understand.

Thank you for pointing this out. It was confusing. We have explained better in the text. 

“The results of the Spearman correlation regarding the UPDRS-III-Factor I and the STAT-ONTM measures have been previously studied. In the study of Rodríguez-Molinero et al, the correlation between the algorithm outputs and the UPDRS-III- Factor I was -0.67 (p < 0.01) [46]”. In this study the comparison was directly between the Factor I and the GF while in our study we have obtained the variation of the mean GF of the same patient in the two monitorizations, which has 0.6. Although these results cannot be directly compared, the correlation obtained in our study shows consistent results with the previous ones. Recently, Pérez-López and collaborators, have found a Spearman correlation of – 0.63 (p> 0.001) between the mean fluidity and the UPDRS-III- Factor I [54].” Page 20, lines 1-10. 

19. p15 L10 - I don't think you have explained the limitations of the study.

Thank you for your comment. We have added this missing information in the text. 

“The present work has several limitations. Due to the design of the study, heterogeneity in both the PD sample and the treatment groups couldn’t have been avoided. Moreover, we haven’t used other specific clinical scales for evaluating the WO, dyskinesia or FoG, nor the Hauser diaries have been used. Finally, the interesting point of the study that is performed in a “free-living-environment” implies that some false positives are unavoidable”.Page 20, lines 1-6. 

20. Finally I will be cheeky enough to ask what the magic feature of machine learning is that warrants the special mentioning on p15 L11? If some other method works, would it be inferior because it is not machine learning? PKG is not based on machine learning and I think there is a reasonable body of evidence to suggest that it can still provide additional value to neurologists in some situations.

Of course, your comment is perfectly fine. Thank you. According to your comment we have corrected in the introduction “Novel devices based either on machine-learning approaches or statistical-based methodology, can be used to capture and monitor motor PD symptoms and MC, or to objectively assess response to dopaminergic therapy [20-33].” Page 4, lines 1-4. 

This is not our will to compare PKG against STAT-ON in this paper (Hence the following text is not in the manuscript). We think they are different sensors and measure different things. For example, wrist-worn devices might not characterize dyskinesia in the trunk or legs or bradykinesia and FoG, but could characterize tremor while STAT-ON can’t. STAT-ON was built with supervised machine learning methods with 92 patients and validating in many more achieving more than 0.9 in sensitivity and specificity in ON-OFF algorithms, bradykinesia, dyskinesia and FoG. The system seems to characterize well motor symptoms and we thought it was worthy to evaluate it in real conditions as PKG has been done. As far as we know only one study has compared PKG and STAT-ON, being quite favorable to STAT-ON, but, as said, there is no will to compare the systems. On the other hand, machine learning, due to its mathematical base generalize better a classification problem. This causes more complexity in mathematical models but, current microprocessors are capable to afford this complexity. As far as we know, and according to Griffiths et al. in 2012, the PKG algorithm is based on 3 features to obtain 2 indexes (BKS and DKS), but no machine learning has been executed, and the generalization could lead to several false positives.

Thank you very much for your comments, suggestions and observations. 

Reviewer #2: Summary

Caballol et al. present an observational study of 29 PD patients who wore a STAT-ON wearable inertial sensor for one week before treatment and one week three months after treatment and 5 PD patients whose treatment did not need adjusting. From the one week of observation, the four days with most time spent in Off were selected for further analysis. Statistically significant changes were seen in the percentage of Off-Time, number of steps and gait fluidity in the group of patients with intervention. The aim of this paper was to show that wearable sensors can be used as a complementary tool to monitor therapy impact and assess PD motor complications. The following aspects may help improve the quality of the manuscript:

General Comments

The authors should thoroughly revise the language and make sure sentences end with full stops to improve readability and comprehension. I have only mentioned some from the first few paragraphs in the minor comments.

We thank the reviewer the comments. We have reviewed the text, introducing more full-stops. 

If the number of steps increased significantly, yet the minutes walking per day did not, this means that the walking speed increased. This could be an interesting aspect for the discussion.

We thank the reviewer to point this out. We have emphasized this observation in the discussion (See Page 16, Lines 20-23). 

Major

Introduction:

-Could you emphasize what the challenges wearable sensor-based systems have and why you believe there is “lack of external validation”? 

Thank you for your comment. We have clarified better this issue in the text as follows: “The main challenge relies on comparing the clinical practice with devices against the clinical practice without devices and seeing if the wearable sensor-based systems can improve PD motor symptoms. So far, only comparisons between sensors and questionnaires have been done, but no clear evidence has been shown yet (See, Page 4, Lines 13-17). 

-Why did the authors focus only on the STAT-ON sensor? More information about wearables the advantages and disadvantages of the data that can be gathered would be useful. 

Thank you for your observation. We have completed more information about this point in the introduction. “Novel devices based either on machine-learning approaches or statistical-based methodology, can be used to capture and monitor motor PD symptoms and MC, or to objectively assess response to dopaminergic therapy [20-33]. The Kinesia 360TM is composed of two sensors, wrist-worn and ankle-worn [32]. It uses a gyroscope, and the algorithms give information regarding tremor, dyskinesia, slowness, mobility, posture and steps. The PKGTM has been extensively used. It is a waist-worn sensor that can detects ON/OFF states, bradykinesia, dyskinesia, tremor and inactivity state but not FoG, gait or falls [26-30]. The PDMonitorTM uses a five-device system and can detect ON/OFF MF, bradykinesia, dyskinesia, tremor, FoG, gait measures and rest state but not falls. Although these novel tools are promising, some challenges such as the lack of external validation could limit their implementation [8]. (See, page 4, lines 3-13). 

Methods:

-Why did the authors select only 4 days with more hours in off-time? Does this not introduce a bias by selecting the days with more off-time hours, especially if the percentage of off-time is compared before and after intervention and is one of the primary outcomes?

Thank you for pointing this out. We have explained better this issue in the text as follows: 

“Although patients were asked to wear the sensor for 12 hours, the time of use during the day was variable. Therefore, all those days with less than 8 hours of monitoring were omitted. The variation of having days with 8 to more than 12 hours of monitoring was minimized using the percentage of time in OFF with respect to the total time monitored. Besides, in our experience there are some patients with fewer hours monitored that put on the sensor later in the morning. This can lead to missing data of the morning-Off’s. The most appropriate and simple criterion to minimize this effect and not introduce bias in the sample was to compare the 4 worst days of each patient in terms of the percentage of OFF with respect to the monitored time.”. See page 8, lines 13-22.

-Given different group sizes, or, rather, difference in variance between groups, the Welch test should be used instead of the t-test.

Thank you for your suggestion. The Welsch test compares two medians in case of normality but with inequality of variances (heteroscedasticity). In our work, since we cannot assume normal distributions due to the small n, we have directly applied non-parametric tests (U Mann Whitney). For this reason, we cannot apply the Welsch test.

-How are variables such as dyskinesia per day, time in on/off, FoG per day calculated? Is there maybe a paper that could be cited where the algorithms from STAT-ON are explained? 

Thank you for your comment. We have now added more information in the text about the sensitivity and specificity of the On/Off, dyskinesia and FoG algorithms in the discussion. See discussion, page 18, lines 5-6, page 19, lines 7-8 and Lines 19-20. 

STAT-ON is quite complex and with several publications in computer science journals. The main STAT-ON publications can be found on: 

Rodríguez-Martín D, Cabestany J, Pérez-López C, Pie M, Calvet J, Samà A, et al. A New Paradigm in Parkinson’s Disease Evaluation With Wearable Medical Devices: A Review of STAT-ONTM. Front Neurol 2022;13. https://doi.org/10.3389/fneur.2022.912343.

Pérez-López C, Samà A, Rodríguez-Martín D, Català A, Cabestany J, Moreno-Arostegui J, et al. Assessing Motor Fluctuations in Parkinson’s Disease Patients Based on a Single Inertial Sensor. Sensors 2016;16:2132. https://doi.org/10.3390/s16122132.

Samà A, Pérez-López C, Rodríguez-Martín D, Català A, Moreno-Aróstegui JM, Cabestany J, et al. Estimating bradykinesia severity in Parkinson’s disease by analysing gait through a waist-worn sensor. Comput Biol Med 2017;84. https://doi.org/10.1016/j.compbiomed.2017.03.020.

Pérez-López C, Samà A, Rodríguez-Martín D, Moreno-Aróstegui JM, Cabestany J, Bayes A, et al. Dopaminergic-induced dyskinesia assessment based on a single belt-worn accelerometer. Artif Intell Med 2016;67. https://doi.org/10.1016/j.artmed.2016.01.001.

Rodríguez-Martín D, Samà A, Pérez-López C, Català A, Moreno Arostegui JM, Cabestany J, et al. Home detection of freezing of gait using support vector machines through a single waist-worn triaxial accelerometer. PLoS One 2017;12:e0171764. https://doi.org/10.1371/journal.pone.0171764.

(All the above references are cited in the text). 

-Since multiple tests were executed, a correction for multiple testing (e.g. Bonferroni) should be employed.

We thank and appreciate your comment. 

Assuming the nature of our study (a pilot study), we have considered our results as exploratory results. In line also with the comments of the reviewer 1, we don’t have applied the multiples test corrections. We hope to publish future works with larger samples and applying the multiples tests. 

Results:

-Pg. 13, lines 3-7: in order to correlate sensor parameters and clinical evaluation parameters, either the mean (or other summary statistic) of the repeated measure (sensor parameter) or repeated measure correlation must be used to test for correlation between the two parameters, otherwise the significance is inflated. Were repeated measures or means per patient for the whole timeframe used? It is unclear from the text.

Thank you for comment. We have clarified better in the text: “The outcome rate of the sensor’s information is provided every 30 minutes. This means that if 12 hours of monitoring are captured per day for 7 days, a total of 168 outcomes are obtained. Obviously, this is not directly comparable with a single clinical evaluation. In order to obtain a single diagnostic data for all the monitoring, different criteria and aggregations have been applied, which we detail below. In our experience, the most natural way to aggregate the sensor information is through a timeframe of one day. This means that the percentage of the different variables (Off, On, Intermediate- Time or dyskinesia) have only been computed if the monitoring period is more than 8h per day. From these daily measurements, we analyse the 4 worst days with higher OFF percentages. Finally, and with the aim of normalising the data amongst the patients, from these obtained 4 days, we calculate the mean of all the parameters. This way, it is obtained in a single data diagnosis which could be comparable to the clinical evaluation carried out in the consultation.” Page 20, lines 1-10. 

· Data Availability: the authors state that “all relevant data are within the manuscript and its Supporting Information files”. Does this mean the sensor data will be uploaded in the supporting information files? I would highly recommend to make the data publicly available by means of a public repository.

Thank you for pointing this out. We have updated our “Data Availability statement”: 

All relevant data are available upon request, due to ethical restrictions imposed by Spanish Agency for Drugs and Medical Devices and the Local Clinical Research Ethics Committee, related to approved consent procedure and protecting privacy. Data can be available upon request to Núria Caballol, email: nuriacaballol@hotmail.com and Àngels Bayés, email: abayes@uparkinson.org. 

Minor

-Authors state that “PD patients wore a validated WIS for one week. WIS data were analyzed before treatment and three months after therapeutic changes.” (Page 2, line 11). Rephrasing would help, as it currently gives the impression that the WIS were worn for one week total. E.g.: “… WIS for one week before treatment and one week three months after therapeutic changes.”

We thank the reviewer to point this out. We have changed the sentence in the abstract, page 2, lines 11-13: “We conducted an observational, open-label study. PD patients wore a validated WIS (STAT-ONTM) for one week, before treatment, and one week three months after therapeutic changes.”

-Please take a look at your figure numbering, in the manuscript you mention Figure 1 c-d, yet they are not present among your figures.

Sorry for the mistake and thank you for pointing this out. We have corrected the Figure 1. We have completed it for a better understanding of the STAT-ON report regarding the information about the percentages of Off-Time and On-Time.

-Table 1: please refrain from expressing percent in parentheses if standard deviation is in parenthesis for other variables.

Thank you. I don’t know if I’m understanding the comment. There were two “%” in the “males” and “females” that have been deleted. Otherwise, we have indicated in the table legend “Values expressed as mean (standard deviation) except when indicated otherwise”. I hope it will be enough

-Page 2, line 14 missing closing parenthesizes and full stop. 

Thank you. Corrected. 

-Line15: Unclear what MF is, as used first time.

Thank you

Corrected. 

-Page 3, line 14: Not sure if “Full-filling” is the right formulation, “Completing” could be better.

Thank you for your comment. It has been corrected. 

We appreciate all your comments and suggestions. Thank you.

---

## [Decision Letter · Decision Letter 1]

19 Dec 2022

Feasibility of a wearable inertial sensor to assess motor complications and treatment in Parkinson’s disease

PONE-D-22-12897R1

Dear Dr. Caballol,

We’re pleased to inform you that your manuscript has been judged scientifically suitable for publication and will be formally accepted for publication once it meets all outstanding technical requirements.

Kind regards,

Keisuke Suzuki, MD, PhD

Academic Editor

PLOS ONE

Additional Editor Comments (optional):

Reviewers' comments:

Reviewer's Responses to Questions

**Comments to the Author**

1. If the authors have adequately addressed your comments raised in a previous round of review and you feel that this manuscript is now acceptable for publication, you may indicate that here to bypass the “Comments to the Author” section, enter your conflict of interest statement in the “Confidential to Editor” section, and submit your "Accept" recommendation.

Reviewer #1: All comments have been addressed

Reviewer #2: All comments have been addressed

2. Is the manuscript technically sound, and do the data support the conclusions?

Reviewer #1: (No Response)

Reviewer #2: Yes

3. Has the statistical analysis been performed appropriately and rigorously? 

Reviewer #1: (No Response)

Reviewer #2: Yes

4. Have the authors made all data underlying the findings in their manuscript fully available?

Reviewer #1: (No Response)

Reviewer #2: No

5. Is the manuscript presented in an intelligible fashion and written in standard English?

Reviewer #1: (No Response)

Reviewer #2: Yes

6. Review Comments to the Author

Reviewer #1: (No Response)

Reviewer #2: Dear editor, thanks for giving me the chance to review this paper again. This is now fine with me.

Yours Sincerely,

Walter Maetzler

7. PLOS authors have the option to publish the peer review history of their article (what does this mean?). If published, this will include your full peer review and any attached files.

Reviewer #1: **Yes: **Filip Bergquist

Reviewer #2: No

---

## [Editor Report · Acceptance letter]

22 Dec 2022

PONE-D-22-12897R1 

Feasibility of a wearable inertial sensor to assess motor complications and treatment in Parkinson’s disease 

Dear Dr. Caballol:

I'm pleased to inform you that your manuscript has been deemed suitable for publication in PLOS ONE. Congratulations! Your manuscript is now with our production department. 

Kind regards, 

on behalf of

Dr. Keisuke Suzuki 

Academic Editor

PLOS ONE